# Evolution of Seed Dispersal Modes in the Orchidaceae: Has the *Vanilla* Mystery Been Solved?

Adam P. Karremans [1,*], Charlotte Watteyn [1,2], Daniela Scaccabarozzi [3,4], Oscar A. Pérez-Escobar [5] and Diego Bogarín [1,6]

1    Lankester Botanical Garden, University of Costa Rica, Cartago P.O. Box 302-7050, Costa Rica; charlotte.watteyn@kuleuven.be (C.W.); diego.bogarin@ucr.ac.cr (D.B.)
2    Department of Earth and Environmental Sciences, KU Leuven, Celestijnenlaan 200E, P.O. Box 2411, 3001 Leuven, Belgium
3    Department of Ecology and Genetics, Uppsala University, 75236 Uppsala, Sweden; daniela.scaccabarozzi@ebc.uu.se
4    School of Molecular and Life Sciences, Curtin University, Bentley 6102, Australia
5    Royal Botanic Gardens, Kew, Richmond TW9 3AE, UK; o.perezescobar@kew.org
6    Evolutionary Ecology Group, Naturalis Biodiversity Center, Darwinweg 2, 2333 CR Leiden, The Netherlands
*    Correspondence: adam.karremans@ucr.ac.cr

**Abstract:** Orchid seeds are predominantly wind-dispersed, often developed within dry, dehiscent fruits that typically release millions of dust-like seeds into the air. Animal-mediated seed dispersal is a lesser-known phenomenon in the family and predominantly occurs in groups belonging to early-diverging lineages bearing indehiscent, fleshy fruits with hard, rounded, dark seeds. In this review, we explore the evolutionary trends of seed dispersal mechanisms in Orchidaceae, focusing on the pantropical genus *Vanilla*. Notably, certain Neotropical species of *Vanilla* produce vanillin-aromatic compounds synthesized naturally in their fruits, which plays a pivotal role in seed dispersal. Ectozoochory occurs in dry, dehiscent fruits, whose seeds are dispersed by (i) male euglossine bees collecting the fruit's vanillin aromatic compounds and (ii) female stingless bees collecting the fruit's mesocarp. Endozoochory occurs in (iii) highly nutritious, indehiscent fruits consumed by terrestrial mammals or (iv) fleshy, dehiscent fruits whose mesocarp is consumed by arboreal mammals. Wind dispersal appears to be a derived state in Orchidaceae and, given its predominance, a trait likely associated with enhanced speciation rates. Zoochory primarily occurs in groups derived from early-diverging lineages; occasional reversions suggest a link between dispersal mode and fruit and seed traits. Interestingly, fruit dehiscence and fleshiness in *Vanilla* lack phylogenetic signal despite their role in determining dispersal modes, suggesting potential environmental adaptability.

**Keywords:** anemochory; birds; insects; mammals; orchids; seed dispersal; vanillin; zoochory





## 1. Background

Vanilla is an economically significant spice of global importance, widely used in a broad spectrum of products offered by the food, cosmetics, and pharmaceutical industries. Vanilla flavoring is derived from vanillin and related aromatic compounds, which are naturally extracted from the fruits, better known as beans or pods, of orchids belonging to the genus *Vanilla* Mill. [1,2]. *Vanilla* orchids grow as climbing vines in the tropical regions of Africa, America, and Asia, where over 120 species are known to occur in the wild [1,3]. However, only one specific clade of *Vanilla* species native to tropical America (*Vanilla* sect. *Xanata*) produces fruits that contain highly esteemed vanillin and related compounds. This includes *Vanilla planifolia* Andrews, the commercially most important species. Although Mexico is generally recognized as the birthplace of vanilla cultivation [4], the leading vanilla-producing countries today are Madagascar (3070 tons/yr) and Indonesia (1456 tons/yr) [5]. Both countries lack the natural pollinators and seed dispersers of vanilla,

as they are located outside the native growing range of all members of *Vanilla* sect. *Xanata*, including *V. planifolia*. Consequently, vanilla plants introduced into these areas do not naturally develop fruits or reproduce sexually by cross-pollination. Therefore, flowers are hand-pollinated, and plants are reproduced through vegetative cloning. As a result, these practices are now widespread among all vanilla-producing countries, even in the Neotropics, because they result in high reproduction rates and satisfying yields [4]. Unfortunately, these practices have also led to a decrease in genetic diversity in the cultivated species of the vanilla crop, which in turn has made vanilla plants highly vulnerable to abiotic and biotic stressors [6–9]. Given that climate change is expected to negatively impact the agricultural sector [10,11], vanilla cultivation practices need to be improved to safeguard the future of this beloved spice and the associated economic activities of this industry.

A possible solution lies In the use of the gene pool available among the wild relatives of the vanilla crop, which includes wild populations of *V. planifolia* but also closely related species with potential traits of interest for crop improvement and breeding [9,12]. Advancements in molecular biology tools have substantially improved our knowledge of phylogenetic relationships, inter- and intra-specific diversity, reproductive systems, and mutualistic interactions within *Vanilla*. Studies range from assessing the genetic variation within *V. planifolia* and closely related species (e.g., [6,13–22]) to enhancing *Vanilla* taxonomy [3,23,24], confirming hybrid progeny and establishing marker-trait associations for crop improvement and breeding (e.g., [22,25–27]). Furthermore, evidence for the presence of diverse mating systems (allogamy vs. autogamy) in *V. planifolia* and related species was provided via molecular studies using isozyme and microsatellite markers [28,29].

Anthropogenic pressures, however, are putting a severe strain on the survival of wild *Vanilla* populations, with most species already having a (critically) endangered status on the IUCN red list [30]. Despite their critical role in species survival, ecological interactions between vanilla plants and their hosts, patrolling insects, pollinators, seed-dispersers, as well as wild seed germination and mycorrhizal associations remain largely undocumented and poorly understood, with numerous knowledge gaps persisting [31–43]. However, unlike most other orchid genera, using character and observational data from extant species suggests a diversity of dispersal modes co-existing within *Vanilla*. Therefore, phylogenetic inferences using current data can reveal novel insights into the evolutionary history of seed dispersal in *Vanilla* and Orchidaceae in general. Seed dispersal is an important stage in the life cycle of a plant, as it influences its reproductive success and, ultimately, survival, thus directly affecting gene frequencies and geographical range of populations [44]; the evolutionary dynamics favoring specific dispersal modes are mostly expected to respond to increased fitness. For centuries, scientists have wondered how vanilla seeds are dispersed and what ecological role vanillin and related aromatic compounds have. The answer—at least part of it—has finally been revealed, and these insights could serve as the foundation for an integral conservation plan across *Vanilla*'s natural distribution range. A recent study by Karremans et al. [42] evidenced a multi-modal seed dispersal mechanism based on the presence of vanillin in species belonging to *Vanilla* sect. *Xanata*, including both ectozoochory and endozoochory. Furthermore, the study showed the occurrence of both dehiscent and indehiscent fruits, even within the same species, and suggests that this fruit trait may be one of the main drivers determining the dispersal mode.

The aim of this review is to summarize the available information on seed dispersal modes within the Orchidaceae, thereby highlighting evolutionary trends and current knowledge gaps. Given the diversity and multimodality of seed dispersal mechanisms in *Vanilla*, combining molecular biology tools with the latest ecological evidence allows us to elucidate patterns in fruit and seed character evolution. Lastly, we identify potential research frontiers, applying molecular biology tools to facilitate evaluation and prediction of seed dispersal modes within *Vanilla* and the Orchidaceae.

## 2. Animal-Mediated Seed Dispersal in Orchidaceae

Orchid seeds are predominantly adapted to wind dispersal since orchid fruits or pods are typically dry and dehisce longitudinally while still attached to the plant [31]. As such, the millions of dust-like seeds can be easily uplifted and carried away by the wind. Nevertheless, there are a few notable exceptions [31,42,45–58] (Table 1; Figure 1). Hydrochory, or the dispersal of seeds by water, is an extremely rare phenomenon that has been documented in only two species within the Epidendroideae subfamily, *Disa uniflora* P.J.Bergius [52,59] and *Epipactis gigantea* Dougl. ex Hook. [52]. These species are typically found near waterbodies, and their fruit and seed traits differ from those of animal-dispersed orchids [52]. Certain orchids—primarily found in early diverging clades—show fleshy fruits that bear hard, rounded, dark seeds. These orchids are known to occur in *Apostasia* Blume and *Neuwiedia* Blume, the two genera of the Apostasioideae subfamily, the genus *Selenipedium* Rchb.f. of the Cypripedioideae subfamily, and in the genera *Cyrtosia* Blume and *Vanilla* of the Vanilloideae subfamily [31,42,46,49–58]. Fleshy fruits and sclerified seeds seem to be very rare in the Orchidoideae and Epidendroideae subfamilies. Among the former, it has only been reported in the genus *Rhizanthella* R.S.Rogers, while in the second, it is known to occur in the genera *Palmorchis* Barb. Rodr. [31,46,49,51,52] and *Yoania* Maxim. [55,56]. In the subfamily Vanilloideae, sclerified seeds are also found in some genera that bear non-fleshy, non-aromatic fruits, such as *Epistephium* Kunth, *Erythrorchis* Blume, *Galeola* Lour., and *Pseudovanilla* Garay. However, the seeds of those orchids are winged and believed to be dispersed by anemochory [48,51]. This may indicate a transitional phase, potentially leading to animal dispersal or a regression to wind dispersal.

**Table 1.** Exceptions to wind dispersal in Orchidaceae.

| Genus [Subfamily] | Species | Seed Dispersal Mechanism | Fleshy/Aromatic Fruit | Sclerified Seed Coat/Winged | Dispersal Agent | References |
|---|---|---|---|---|---|---|
| *Apostasia* [Apostasioideae] | | Zoochory | Yes/most | Yes/No | Crickets | [49,50,52] |
| *Apostasia* [Apostasioideae] | *A. nipponica* | Zoochory | Yes/Yes | Yes/No | Crickets | [57] |
| *Neuwiedia* [Apostasioideae] | *N. grifithii* | Anemochory? | No/No | Yes/No | Wind? | [49,50,52] |
| *Neuwiedia* [Apostasioideae] | *N. veratrifolia* | Anemochory? | No/No | Yes/No | Wind? | [49,50,52] |
| *Neuwiedia* [Apostasioideae] | *N. borneensis* | Zoochory | Yes/Yes | Yes/No | Birds? | [49,50,52] |
| *Neuwiedia* [Apostasioideae] | *N. singapureana* | Zoochory | Yes/Yes | Yes/No | Birds | [58] |
| *Neuwiedia* [Apostasioideae] | *N. zolingeri* var. *javanica* | Zoochory | Yes/Yes | Yes/No | Birds? | [49,50,52] |
| *Selenipedium* [Cypripedioideae] | *S. chica* | Zoochory? | Yes/Yes | Yes/No | Unknown animals | [31,46,49,51] |
| *Clematepistephium* [Vanilloideae] | | Anemochory? | No/No | No?/Yes | Wind? | [48,52] |
| *Cyrtosia* [Vanilloideae] | | Zoochory | Yes/Yes | Yes/No | Birds | [53] |
| *Cyrtosia* [Vanilloideae] | *C. septentrionalis* | Zoochory | Yes/Yes | Yes/No | Frugivorous birds | [48,52,54] |
| *Epistephium* [Vanilloideae] | | Anemochory? | No/No | Yes/Yes | Wind? | [45,48,52] |
| *Erythrorchis* [Vanilloideae] | | Anemochory? | No/No | Yes/Yes | Wind? | [48,52] |
| *Eriaxis* [Vanilloideae] | | Anemochory? | No/No | No?/Yes | Wind? | [48,52] |
| *Galeola* [Vanilloideae] | | Anemochory? | No/No | Yes/Yes | Wind? | [45,48,52] |
| *Pseudovanilla* [Vanilloideae] | | Anemochory? | No/No | Yes/Yes | Wind? | [48,52] |
| *Vanilla* [Vanilloideae] | | Zoochory | Yes/Yes | Yes/No | Bees, Crickets, Mammals | [42] |
| *Rhizanthella* [Orchidoideae] | | Zoochory? | Yes/Yes | Yes/No | Mammals? | [47,52] |
| *Palmorchis* [Epidendroideae] | | Zoochory? | Yes/Yes | Yes/No | Unknown animals | [31,46,49,51,52] |
| *Yoania* [Epidendroideae] | *Y. amagiensis* | Zoochory | Yes/Yes | Yes/No | Crickets | [55,56] |

**Table 1.** *Cont.*

| Genus [Subfamily] | Species | Seed Dispersal Mechanism | Fleshy/Aromatic Fruit | Sclerified Seed Coat/Winged | Dispersal Agent | References |
|---|---|---|---|---|---|---|
| *Yoania* [Epidendroideae] | *Y. japonica* | Zoochory | Yes/Yes | Yes/No | Crickets | [55,56] |
| *Disa* [Epidendroideae] | *D. uniflora* | Hydrochory | No/No | No/No | Streams | [52,59] |
| *Epipactis* [Epidendroideae] | *E. gigantea* | Hydrochory | No/No | No/No | Streams | [52] |

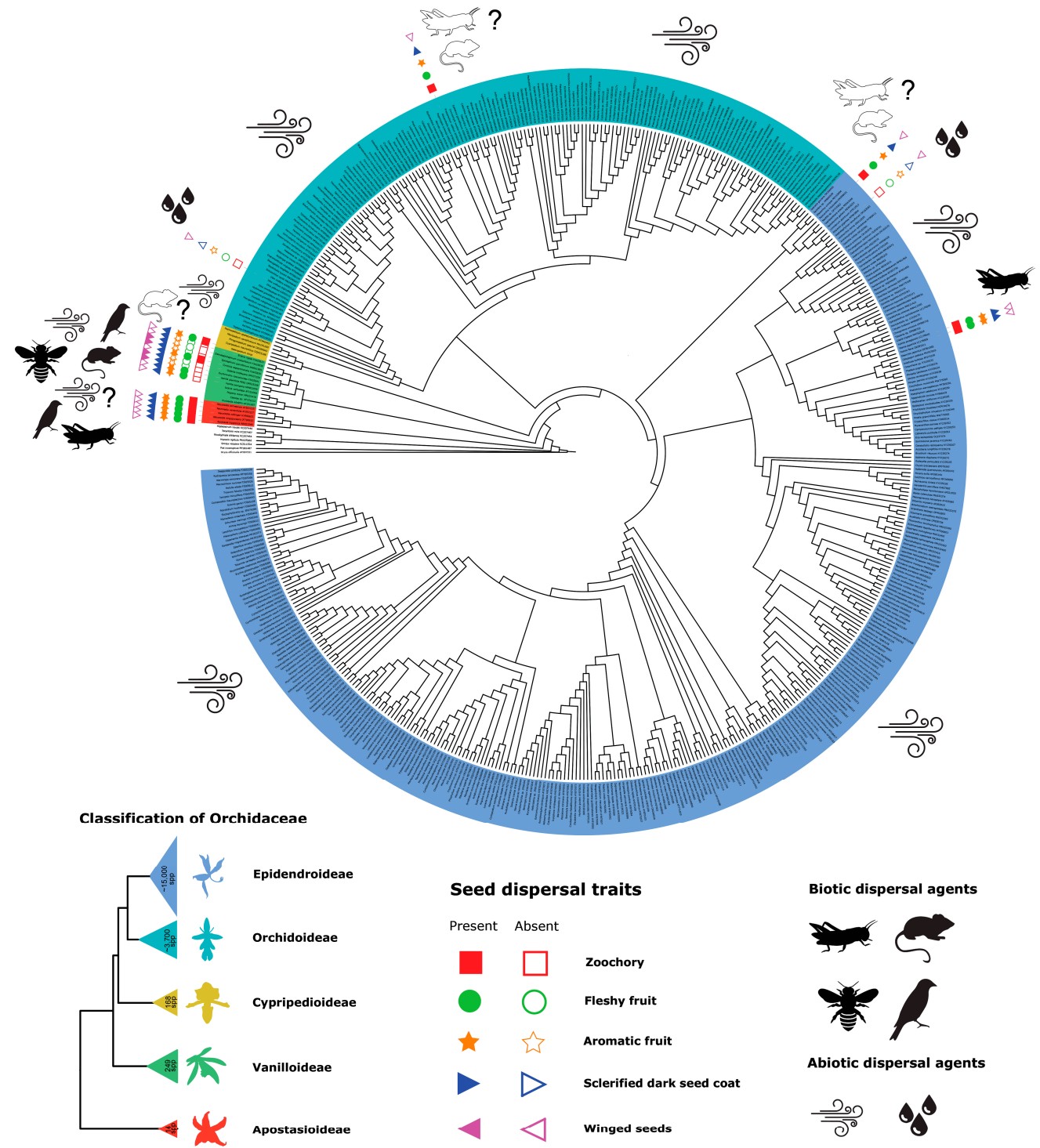

**Figure 1.** Evolution of seed dispersal mechanisms and fruit traits in the Orchidaceae.

The fleshy, often aromatic fruits containing sclerified seeds are suggested to employ animal vectors rather than being adapted to wind dispersal, and this has indeed been proven in certain cases. For example, in tropical Asia, avian dispersal has been observed in *Cyrtosia* [54] and *Neuwiedia* [58], while seed dispersal by crickets was shown to occur in *Apostasia* [57] and *Yoania* (*Y. amagiensis* Nakai & Maek. and *Y. japonica* Maxim.) [55,56]. Both *Cyrtosia septentrionalis* (Rchb.f.) Garay and *Neuwiedia singapureana* (Wall. ex Baker) Rolfe have fleshy fruits that turn from green to bright red as they mature, a color that has been shown to attract birds [60,61]. Fruits of *Apostasia* and *Yoania* are eaten by crickets and camel crickets and display green and pinkish-white colors, respectively [55–57]. In these examples, the dark, rounded seeds were recovered from the feces of animals that fed on the fleshy fruits, and germination trials showed that the seeds are viable after passing the digestive system, confirming the presence of zoochory among these orchid genera.

Zoochory is suggested to be the main dispersal mechanism in *Vanilla*, and no other orchid genus seems to have such diverse dispersal modes and agents as this large pantropical group. Indeed, unlike the tiny, dust-like, transparent seeds of most other wind-dispersed orchids, like other zoochorous orchids, *Vanilla* seeds are relatively heavy, rounded, and covered by a hard, black seed coat. Moreover, seeds of certain *Vanilla* species are packed into a notoriously vanillin-fragrant fruit. This fragrance has been suspected to play a role in the dispersal of *Vanilla* seeds, but evidence for this important ecological function has been absent until very recently. Animals ranging from ants and bats, bees, crickets and reptiles, birds, rodents, marsupials, and even monkeys have been reported in the literature as possible *Vanilla* seed dispersers. Fruit and seed traits, therefore, are suggested to result from the interplay of abiotic factors, such as habitat, and biotic factors, in the sense of dispersing agents.

## 3. Animal-Mediated Seed Dispersal in *Vanilla*

### 3.1. Vanilla Fruit Trait Diversity

Bouetard et al. [62] noted that vanillin-bearing species, so-called 'fragrant' *Vanilla*, come exclusively from the Neotropics and belong to a single clade: *Vanilla* sect. *Xanata*. Except for a few early diverging species (e.g., *V. bicolor* Lindl., *V. palmarum* Lindl.), most other members of this section seem to bear fruits that contain vanillin and related compounds. Hereafter, we will refer to these as the fragrant *Vanilla*. This clade is sister to *Vanilla* sect. *Tethya* Soto Arenas & P.J.Cribb, which includes the Old World *Vanilla* species found in tropical Asia and Africa and the leafless species from the Caribbean. Together, these two sections comprise *Vanilla* subgen. *Xanata* Soto Arenas & P.J.Cribb, which in turn is sister to the exclusively Neotropical *Vanilla* subgen. *Vanilla*, known as the 'membranaceous' group. As far as is known, both *Vanilla* subgen. *Vanilla* and *Vanilla* sect. *Tethya* consistently lack vanillin relatives in their fruits, and we will hereafter refer to these as the non-aromatic *Vanilla*. However, we emphasize that the lack of vanillin and related compounds should not be confused with a complete lack of odor. Species belonging to *Vanilla* sect. *Tethya* produce a sweet aroma and members of *Vanilla* subgen. *Vanilla* have fruits with a rather grassy, slightly sweet, fermented aroma. The non-aromatic members of *Vanilla* sect. *Tethya* and *Vanilla* subgen. *Vanilla* are, therefore, not completely odorless; their fruits emit a slightly sweet, fermented smell that is likely appealing to certain animals.

Another important and often neglected fruit trait that shows variation among *Vanilla* species is fruit dehiscence. Dehiscent fruits split open successively while they mature on the vine, with two valves recoiling from the apex to the base. Indehiscent fruits do not split at all. Instead, they turn yellow and drop to the ground, where they finish maturing. Unfortunately, this fruit trait is rarely noted in the literature and needs to be better understood. Among the fragrant species, *Vanilla hartii* Rolfe, *V. insignis* Ames, *V. odorata* C.Presl., and *V. trigonocarpa* Hoehne, for example, have dehiscent fruits [1]. Interestingly, the dehiscent fruits of *V. hartii* and *V. odorata* are rather non-fleshy and expose their seeds when still attached to the vine. In contrast, the dehiscent fruits of *V. trigonocarpa* are fleshy, exposing its thick pulp as the dorsal valve of the mature fruit recoils. Dehiscence

has also been described in *Vanilla bahiana* Hoehne (a taxonomical synonym of *V. phaeantha* Rchb.f.) and *V. labellopapillata* A.K.Koch, Fraga, J.U. Santos & Ilk.-Borg. [63,64], as well as in *V. cristato-callosa* Hoehne, *V. karen-christianae* Karremans & P.Lehm. (as *V. riberoi* Hoehne), *V. palmarum* Lindl. and *V. pompona* Schiede [33]. Interestingly, unlike the dehiscent Peruvian *V. pompona* populations, Costa Rican populations produce indehiscent fruits that drop to the ground before turning brown, soft, and fragrant [42]. Even though these two populations are sometimes treated as different at the species or subspecies level, they are nevertheless close sisters [3,19]. The same occurs within *V. planifolia*, where the Mexican and commercial cultivars are dehiscent, while plants native to Costa Rica, forming a sister clade [19,21], have indehiscent fruits. Indehiscence has also been observed in *V. dressleri* and the commercial hybrid *Vanilla* × *tahitensis*. The latter is particularly curious, given that both putative parents (*V. odorata* and *V. planifolia*) have dehiscent fruits.

Based on our phylogenetic reconstruction combined with dispersal features of fragrant *Vanilla* species (Figure 2), dehiscence appears to be more common and is likely to be the ancestral condition [65]. Recent evolutionary studies in other plant families suggest that indehiscent fruits evolved from dehiscent ancestors (e.g., Brassicaceae) [66–68]. Curiously, fruit dehiscence (and dispersal modes) appear not to be phylogenetically conserved, and both dehiscence and indehiscence may occur in different populations of a single species. However, within the same population or individual, this fruit trait seems to be invariable. So, the question now arises: has indehiscence evolved several times independently among the fragrant *Vanilla* species, or is gene expression involved in fruit dehiscence plastic and, therefore, expressed in some populations of a single species and not in others?

*3.2. Linking Vanilla Fruit Traits with Dispersal Modes*

3.2.1. Evidence for Ectozoochory

Vanillin, which is found on the inner side of the *Vanilla* fruit, attracts euglossine bees (Apidae: Euglossini)—known as orchid bees for their importance in the pollination of many large-flowered Neotropical orchids. Male euglossine bees search and collect fragrances from diverse sources to concoct a perfume they use for courtship. Among other plant resources, they are known to actively collect the aromatic compounds produced by *Vanilla* fruits. In Mexico, Lozano Rodríguez et al. [69] found that the euglossine bees *Euglossa hemichlora*, *Eug. variabilis*, *Eulaemaexicanta*, *Eul. Polychroma,* and *Exaerete frontalis* visited the mature, dehiscent fruits of *Vanilla planifolia*. *Eulaemaexicanta* was also recorded visiting the dehiscent fruits of *Vanilla odorata* in Mexico [70] and *V. pompona* in Peru [32]. Furthermore, euglossine males demonstrated typical fragrance collection behavior when visiting the dehiscent fruits of *Vanilla cristato-callosa* and *V. pompona* in Peru (Householder et al. [33]). A broad study by Karremans et al. [42] showed that numerous Euglossini species visit the dehiscent fruits of *Vanilla planifolia* and *V. odorata* in Costa Rica. Males of fifteen different species were identified: *Eufriesea surinamensis, Euglossa allostricta, Eug. asarophora, Eug. bursigera, Eug. cybelia, Eug. dodsoni, Eug. hansonii, Eug. heterostricta, Eug. ignita, Eug.exicantea, Eug. tridentata, Eug. villosiventris, Eulaemaexicanta, Eul. meriana* and *Eul.exicantea*. Of these, *Eug. bursigera, Eug. ignita*, *Eug. tridentata,* and *Eul.exicanta* were confirmed to remove seeds of both *V. planifolia* and *V. odorata*, while *Eul.exicantea* removed seeds of *V. planifolia*. In total, about 10% of all Euglossini species, belonging to four out of the five genera, have now been observed visiting *Vanilla* fruits, with several bee species visiting more than one *Vanilla* species. One common and widespread species, *Eul.exicanta*, has been observed visiting *V. odorata*, *V. planifolia,* and *V. pompona* in Mexico, Costa Rica, and Peru. This suggests that there is broad interest from Euglossini in fragrant *Vanilla* fruits of different species across a wide geographical range. Given that fragrance collection is a common behavioral trait among these bees and that the dispersal mechanism is not size-dependent, many more Euglossini species are expected to be able to disperse *Vanilla* seeds.

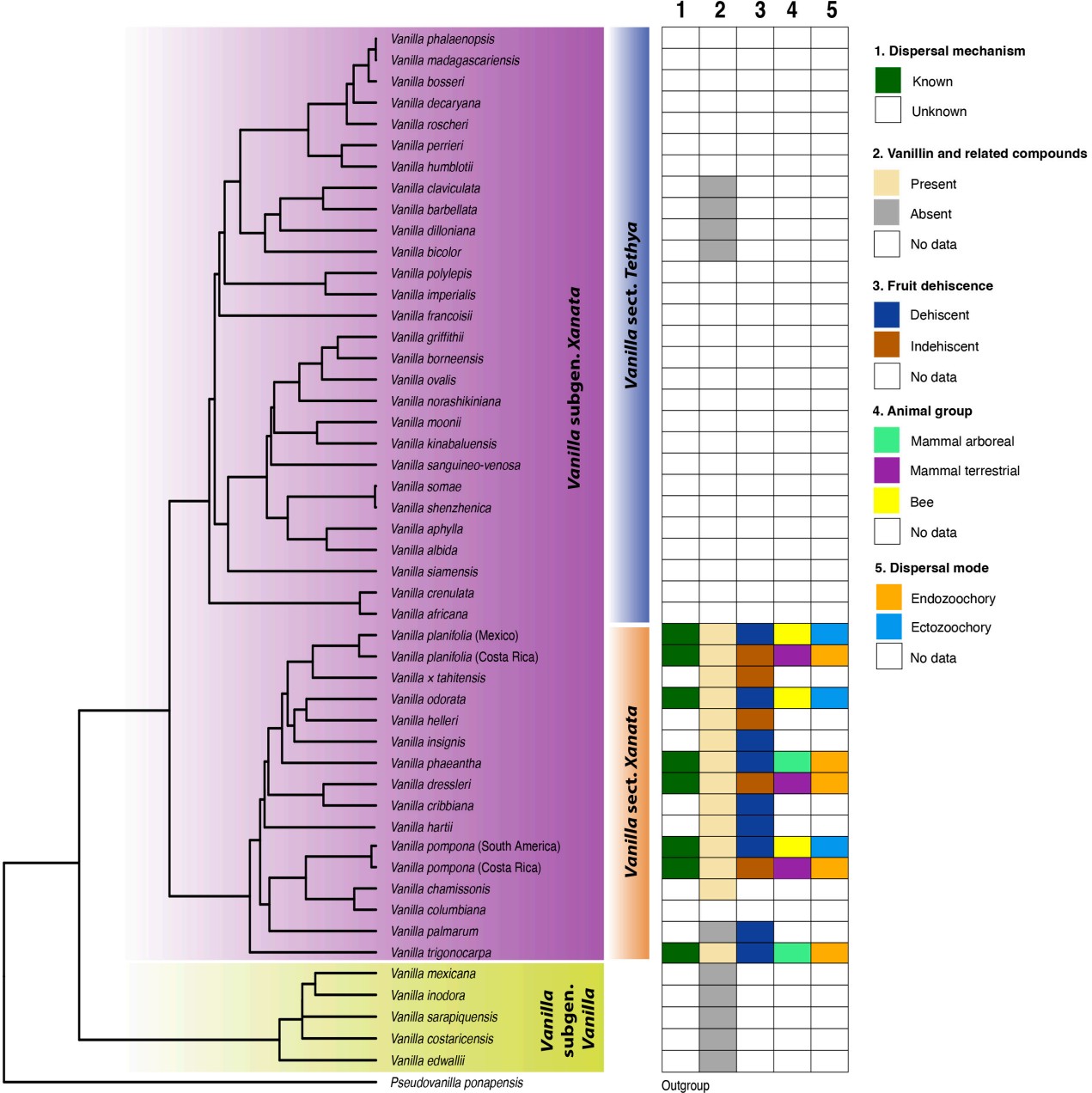

**Figure 2.** Maximum likelihood phylogenetic *Vanilla* tree based on nrITS dataset linked with compiled data on known dispersal modes and fruit traits within *Vanilla*.

Euglossine males are not the only bees that show interest in *Vanilla* fruits. Householder et al. [33], for example, suggested that a species of stingless bee of the genus *Trigona* could be responsible for collecting seeds from the dehiscent fruits of *Vanilla pompona* in Peru. In Costa Rica, Karremans et al. [42] observed female stingless bees (Apidae: Meliponini) collecting the fruit's seed-rich pulp of *V. odorata* and, especially, cultivated *V. planifolia*. More specifically, female bees belonging to the genera *Trigona* and *Scaptotrigona* were seen to actively collect the seeds and store them in specialized pouches in their hind legs, together with other nest-building materials. Recent studies in the Neotropics have shown that stingless bees are indeed attracted to, and may actively collect, vanillin [71,72]. The fact that seeds of *V. odorata* and *V. planifolia* were found on the hind legs of generalist foragers *Trigona fulviventris* and *Scaptotrigona subobscuripennis*, respectively, suggests that other stingless bee species may also visit the fruits and potentially remove and disperse seeds while

collecting the fruit's pulp. Such is likely to be the case for dehiscent *V. pompona* fruits as well, confirming Householder et al.'s [33] observations in Peru. Seed dispersal by bees, or melittochory, is an extremely rare mutualistic relationship. It is known to occur in only three rainforest tree species: *Corymbia torelliana* (F. Muell.) K.D.Hill & L.A.S.Johnson (Myrtaceae), *Coussapoa asperifolia* subsp. *magnifolia* (Trécul) Akkermans & C.C.Berg (Urticaeae), and *Zygiaexicana* (Ducke) Barneby & J.W.Grimes (Fabaceae). The seeds of these tree species are dispersed by Meliponini bees that collect the resin inside the fruits [73–76]. The genus *Vanilla* is the first among the monocots with confirmed seed dispersal by bees, making Orchidaceae only the fourth melittochorous family among plants.

### 3.2.2. Evidence for Endozoochory

*Vanilla* seed dispersal has long been suggested to be mediated by large animals through endozoochory. Unfortunately, most reports have been anecdotal, and none have shown any proof of animals manipulating *Vanilla* fruits until recently, let alone ingesting and defecating the seeds. The diversity of animals that have been suspected to be involved in the dispersal of *Vanilla* seeds includes birds [31,49], reptiles [77], and diverse mammals, such as monkeys [78], marsupials [33], and bats [1,79]. Yet, the subject has remained rather cryptic, lacking resolution as to which and how animals truly interact with fragrant *Vanilla* fruits and the consequences of these interactions.

This changed when the first conclusive evidence of rodents consuming the mature, indehiscent fruits of *Vanilla pompona* was presented by Karremans et al. [35–37]. Videos obtained from motion-activated camera traps placed in several *V. pompona* populations naturally occurring in the Southern Pacific of Costa Rica displayed the Central American spiny rat, *Proechimys semispinosus* (Rodentia: Echimyidae), and Central American agouti *Dasyprocta punctata* (Rodentia: Dasyproctidae) actively handling the indehiscent fruits before consuming them entirely. Karremans et al. [42] further expanded these observations with additional evidence of the rodents *Proechimys semispinosus* and *Sigmodontomys alfari* (Rodentia: Cricetidae), and the marsupial *Didelphis marsupialis* (Didelphimorphia: Didelphidae) consuming the indehiscent fruits of the native *Vanilla planifolia* growing on the Caribbean side of Costa Rica. The study not only provided clear video and photographic proof of the mammals consuming fruits of these two *Vanilla* species in their natural habitat, but intact *V. planifolia* seeds were recovered from the feces of both the spiny rat (*P. semispinosus*) and common opossum (*D. marsupialis*), showing that seeds passed through their digestive systems. Furthermore, in situ and ex situ experiments demonstrate that passing through the mammal's digestive system is not harmful to the seeds but is also not a strict requirement for germination. Protocorms of *V. planifolia* were shown to develop 3.5 months after packets with fecal material containing seeds were placed in the soil at independent locations within the natural populations in Costa Rica. Our preliminary observations on the fleshy, fragrant, indehiscent fruits of *V. dressleri* suggest that this species also attracts rodents that consume the fruits.

Virtually simultaneous observations of animals interacting with *Vanilla* fruits were made in Brazil. A study by Pansarin [38] found that birds interacted with the fruits of multiple *Vanilla* species at the campus of the University of São Paulo in southeastern Brazil. Unfortunately, the fruits of the different *Vanilla* species were displayed simultaneously, in unequal proportions, in an unnatural disposition, and in an intervened environment outside their natural habitat. The mix of mature and immature fruits, belonging to both fragrant and non-fragrant species, simultaneously offered on an artificial platform, discounts any individual attractiveness. Moreover, it disregards the possibility of a magnet effect of one or more particular *Vanilla* fruits, as well as prior-learned food cues and the exploratory behavior of birds given the presentation. It is, therefore, impossible to conclusively say which, if any, of the fruits of the *Vanilla* species tested actually attract birds and if the same occurs under natural conditions in a wild setting. Consumption and passing of the seeds were not shown for any of the *Vanilla* and bird species tested. In contrast, motion-activated camera traps placed at several locations in Costa Rica showed several bird species,

belonging to multiple families, passing close to or over the highly fragrant, mature fruits of *V. odorata*, *V. planifolia,* and *V. pompona* without ever attempting to consume or even reacting to the presence of the fruits [42]. The video footage shows that the fruits of these three species do not elicit any response from birds. It is noteworthy, however, that *Vanilla palmarum*, which differs by having non-fragrant, compact fruits, showed the highest visitation by birds in southeastern Brazil [38]. Therefore, it would certainly be worthwhile to separately re-evaluate if the fruits of that species, and any other non-aromatic *Vanilla*, do, in fact, attract birds or other animals in their natural habitat and if seeds are truly consumed and dispersed through these means.

A follow-up paper by Pansarin and Suetsugu [40] provided evidence that the mature, aromatic, dehiscent fruits of *Vanilla bahiana* (=*V. phaeantha*) are visited by birds, marsupials, and rodents in a semi-natural setting. The bird species were said to be mainly *Mimus saturninus* (Passeriformes: Mimidae) without any further specification. Unfortunately, no evidence was provided regarding the birds' interaction with the fruits. In contrast, the arboreal marsupial *Gracilinanus agilis* (Didelphimorphia: Didelphidae) and the rodent *Oligoryzomys nigripes* (Rodentia: Cricetidae) were documented, using camera traps, actively engaging with, and nibbling on the fruits. The interest from these mammals in the mature *Vanilla* fruits is consistent with aforementioned studies of mammals consuming fragrant fruits and dispersing the seeds. Contrary to the indehiscent *V. planifolia* and *V. pompona*, the dehiscent fruits of *V. phaeantha* do not fall to the ground as they mature, and it was suggested that the observed animals solely consume the exposed pulp without chewing the mesocarp. In southeastern Costa Rica, the arboreal rodent *Nyctomys sumichrasti* (Rodentia: Cricetidae) was recorded consuming the pulp of *Vanilla trigonocarpa* (Karremans et al., pers. obs.), a species that, like *V. phaeantha*, has fleshy, fragrant, dehiscent fruits that are persistent on the vine. Bite and puncture marks on another mature fruit of *V. trigonocarpa*, which had the pulp removed, were identified as being made by bats. Moreover, another arboreal mammal, *Potos flavus* (Carnivora: Procynoidae), was recently recorded consuming mature fruits of *V. planifolia* at Cahuita National Park in Costa Rica (Karremans et al., pers. obs.). Even though ingestion and passing of the seeds remains to be shown in these cases, *G. agilis*, *N. sumichrasti*, *O. nigripes*, and *P. flavus* are all fruit-eating arboreal animals that have been shown to disperse seeds. These observations indicate that seed dispersal through direct pulp consumption by mammals is likely another endozoochory mechanism in *Vanilla*. Claims by Pansarin and Suetsugu [40] that the mesocarp of fragrant *Vanilla* species is generally toxic to mammals are probably premature given that, at least for *Vanilla planifolia* and *V. pompona*, several mammal species have shown to be rather keen on consuming the entire fruit. The authors' own video published with the supporting information shows a *Gracilinanus agilis* individual ripping a piece of the mesocarp of *Vanilla phaeantha* and evidently chewing it. Whether palatability changes with the fruit's maturity is worth investigating further.

Current data show a possible correlation between certain fruit traits and the dispersal agent or mode in *Vanilla* (Figure 2). For example, indehiscent fruits (e.g., *Vanilla dressleri*, *V. pompona*) are dispersed by terrestrial mammals, which consume the mature, nutritious fruits as they fall onto the ground (endozoochory). Dehiscent fruits (e.g., *Vanilla odorata*, *V. phaeantha*) are dispersed by bees, which displace seeds while collecting fragrances or nest-building materials from the fruits (ectozoochory) or arboreal mammals that consume the fruit's pulp (endozoochory). However, the question remains, which fruit traits trigger the attraction of different animal groups (bees vs. arboreal mammals) to dehiscent fruits. Fruit fleshiness and richness of the pulp possibly play an important role therein. The rather dry fruits of *V. odorata* attract bees, while the succulent fruits of *V. phaeantha* and *V. trigonocarpa* attract arboreal mammals that consume the rich pulp. However, only bees have been recorded visiting the dehiscent, fleshy fruits of *V. pompona* (Peruvian populations). The attraction of different animal groups may also depend on the concentration and composition of aromatic compounds in each fruit, which differs significantly among *Vanilla* species and can vary even within a single species [80]. Our understanding of these interactions

is still too partial to provide a clear overview of the mysterious ecology of *Vanilla* seed dispersal. Future in-depth studies with robust field experimentation are needed to explore (i) the relationships between various *Vanilla* fruit traits and corresponding dispersal modes and (ii) the genetic and ecological triggers that promote certain traits and dispersal modes above others.

### 3.3. Flexible Dispersal Modes in Vanilla: A Gene–Environment-Induced Regulation?

An intriguing question that arises is why certain fruit traits and seed dispersal modes vary among populations within a single species. For example, *V. planifolia* exhibits indehiscent fruits in Costa Rica but dehiscent fruits in Mexico, thereby attracting distinct dispersal agents. This variation leads to a shift in dispersal modes, namely a transition from endozoochory to ectozoochory or vice versa. The same holds for *V. pompona*, which produces indehiscent fruits in Costa Rica but dehiscent fruits in Peru. The coexistence of both dehiscent and indehiscent fruits, and, thus, varying dispersal modes and agents within the same species, suggests the possibility of a shift between dispersal strategies in response to, for example, spatiotemporal environmental variability. As previously argued, fruit dehiscence is a critical characteristic that influences seed maturation and dispersal in Brassicaceae [81,82], and specific environmental factors may influence genes regulating fruit (in)dehiscence. For instance, fruit dehiscence has been shown to be regulated by temperature via specific thermosensory activation pathways [83].

Based on our preliminary observations of varying dispersal modes, we see great potential in future studies that provide insights into (i) the molecular and genetic mechanisms underlying the evolutionary transition between specific fruit traits (e.g., dehiscent to indehiscent fruits or vice versa) and (ii) the responses of genes regulating these particular fruit characteristics to (local) environmental conditions, such as climate, light intensity, or habitat type, amongst others. Considering that genes regulating fruit dehiscence and other fruit traits have been successfully studied in several plant species [81,82,84,85] and epidendrioid orchids [86,87], we believe that applying similar molecular approaches will greatly advance our understanding of seed dispersal mechanisms within *Vanilla*. For this, whole-genome and transcriptome sequencing combined with resequencing technologies could be applied to identify and isolate candidate genes responsible for the expression of fruit traits affecting fruit ripening and, thus, seed dispersal. Later studies could then track down these genes using molecular markers to reveal their presence in *Vanilla* species, specifically at the population level. Furthermore, experimental evaluations within controlled settings combined with transcriptome analyses could reveal certain interactions between these candidate genes and specific environmental conditions. This could be especially of interest given climate change causing potential shifts in dispersal modes.

### 4. Conclusions

Orchid fruits and seeds are mainly adapted to anemochory, and this derived state has likely been an important driver of speciation within the family. However, zoochory occurs primarily in groups belonging to the early diverging clades of the family. This ancestral state is often linked to morphological features, including fruit indehiscence and fleshiness and darkening and sclerification of the seed coat. Fruit and seed features also appear to determine dispersal modes within *Vanilla*, the largest and most diverse zoochorous genus in the family. *Vanilla* joins *Apostasia*, *Cyrtosia*, *Neuwiedia*, and *Yoania* as the only orchid genera in which animal dispersal has been proven beyond doubt. Despite lacking a phylogenetic signal for fruit traits within *Vanilla*, fruit fragrance, dehiscence, and fleshiness play a role in determining the attraction to particular animals and, therefore, the dispersal modes. Fragrant *Vanilla* species present multi-modal seed dispersal mechanisms that exploit different behaviors of several groups of animals (Figure 3) and are likely to depend on vanillin and related aromatic compounds to attract seed-dispersing animals. The mature, slender, dehiscent fruits of *Vanilla odorata, V. planifolia* (Mexico), and *V. pompona* (Peru) attract male euglossine bees and female stingless bees that displace or collect seeds

as they collect fragrances and nest-building materials from the fruits, respectively. As such, *Vanilla* represents the first case of plants having their seeds dispersed through fragrance collection, with the above-mentioned species being the only monocot species known to date to have their seeds dispersed through melittochory. Together with *Corymbia torelliana* (Myrtaceae), *Coussapoa asperifolia* subsp. *magnifolia* (Urticaeae), and *Zygiaexicana* (Fabaceae), they are the only known plant species sharing this exceedingly rare mutualistic relationship.

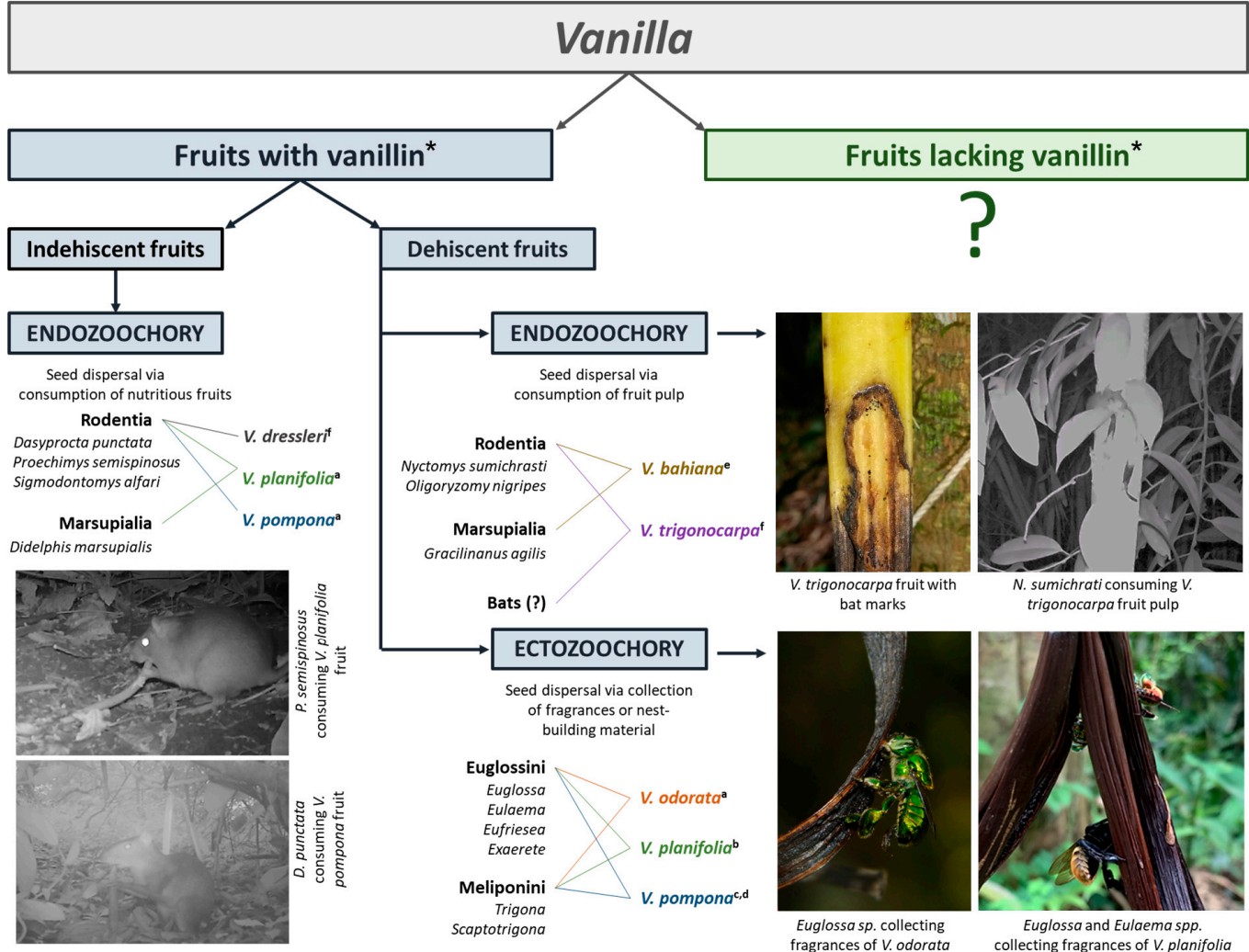

**Figure 3.** Dispersal mechanisms in the orchid genus *Vanilla*. © Adam Karremans. [a] Karremans et al. [42]. [b] Lozano Rodríguez et al. [69]. [c] Lubinsky et al. [32]. [d] Householder et al. [33]. [e] Pansarin and Suetsugu [40]. [f] Karremans et al., pers. obs. * We refer to both vanillin and its related compounds.

The indehiscent fruits of *V. dressleri*, *V. planifolia* (Costa Rica), and *V. pompona* (Costa Rica) are unique among the Orchidaceae in being highly nutritious and attracting terrestrial mammals, especially rodents, that consume the whole fruit and can pass the entire seeds through their digestive system. Mammals also visit and consume the dehiscent fruits of *V. phaeantha* and *V. trigonocarpa*. In this case, the fleshy, persistent fruit is visited by arboreal or flying mammals that mainly consume the pulp and presumably disperse the seeds after passing them through their digestive system. As such, there may be multiple other mammals involved in seed dispersal across the *Vanilla* genus. Secondary dispersal by other animals is surely a possibility in certain cases, too. Crickets, which have been shown to disperse the seeds of other orchids, were found by Karremans et al. [42] feeding upon the indehiscent fruits of *V. planifolia* that had been left out in the field. The crickets, which

belong to a species in genus *Idiarthron* (Tettigoniidae), passed the *Vanilla* seeds intact and could also act as effective dispersers.

The identity of seed dispersers for *Vanilla* species that lack vanillin remains enigmatic. Given the diversity of seed dispersers and mechanisms among the fragrant, New World members of *Vanilla*, one may speculate that numerous interactions involving different animal groups may exist across the broad geographical range of this genus. Birds, which have been observed visiting *Vanilla* fruits, may potentially consume and disperse the seeds of certain *Vanilla* species, as they do for other orchids. However, this remains somewhat speculative. Due to their underdeveloped sense of smell, birds primarily rely on visual cues to locate the fleshy fruits they consume, showing a particular attraction to reddish colors while often ignoring greens. This raises controversy regarding their role in seed dispersal of *Vanilla* species with fragrant, but visually unappealing, fruits. However, it does not exclude them from being potential *Vanilla* seed dispersers. The fleshy, compact, non-fragrant fruits of the greenish *V. palmarum* and the reddish *V. exicantea* and their relatives, for example, are good candidates for dispersal by birds or other groups of animals.

Significant progress has been made recently toward understanding the ecological interactions involving the dispersal mechanisms of *Vanilla* orchids. However, much remains to be studied, particularly under natural conditions within their native habitats. Robust field experimentation is crucial in understanding the intricate details of these complex relationships and how they may have evolved. Furthermore, detailed anatomical and morphological observations of fruits and seeds from groups potentially dispersed by animals are essential to explore the evolution of these traits and their role in seed dispersal. These traits are often underrepresented in the literature and require more comprehensive documentation. Additionally, insights into the genetic basis of these multidimensional dispersal modes could reveal potential adaptations of *Vanilla* species and their populations to specific environmental conditions. Although describing this genetic basis may be highly challenging, we believe that a combination of molecular tools could provide great potential to study these underlying mechanisms. We eagerly expect upcoming and exciting advances in *Vanilla* ecology to further expand our knowledge on fundamental biological aspects and to allow the development of a proper conservation plan for this commercially very important orchid genus.

**Author Contributions:** Conceptualization, A.P.K. and C.W.; methodology, D.B. and O.A.P.-E.; software, D.B. and O.A.P.-E.; formal analysis, D.B. and O.A.P.-E.; investigation, A.P.K., C.W., D.S., D.B. and O.A.P.-E.; data curation, D.B. and O.A.P.-E.; writing—original draft preparation, A.P.K., C.W. and D.S.; writing—review and editing, A.P.K., C.W. and D.S. All authors have read and agreed to the published version of the manuscript.

**Funding:** This research was funded by the Vicerrectory of Research of the University of Costa Rica under project numbers 814-C0-049 and 814-C3-464.

**Data Availability Statement:** No new data has been created for the current review.

**Acknowledgments:** We are very thankful to the staff at JBL and CIBET, Cahuita National Park, Tirimbina Biological Reserve, Las Brisas Nature Reserve, Piro Biological Station, and Finca Christina. The Costa Rican Ministry of Environment and Energy (MINAE) and its National System of Conservation Areas (SINAC) kindly provided the permits and access to protected areas.

**Conflicts of Interest:** The authors declare no conflict of interest.

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
