# Peer review of "Evolution of Seed Dispersal Modes in the Orchidaceae: Has the Vanilla Mystery Been Solved?"

_horticulturae, doi:10.3390/horticulturae9121270_

Round 1

Reviewer 1 Report

Comments and Suggestions for Authors

This is a highly useful and comprehensive review for a very interesting topic in orchid biology by the authors who have recently made significant contributions in this field. I have only minor rewording suggestions as listed below.

Line 46: Unnecessary parenthesis after [4]

Line 62: traits

Line 66 Parenthesis after [6,13-22]

Line 85: respond

Line198: conserved, rather than linked

Figure 1: The illustration of camel cricket does not really look like crickets. Better be redrawn.

Figure 2: More useful to have subgenus and section names on the tree. Presence/Absence for “2. Vanillin and related compounds” should be Present/Absent.

Line 241: Species or taxonomic groups do not “learn”, so the sentence in my opinion can be omitted.

Line 259: shown

Line 269 and 270: Unnecessary periods

Line 340: terrestrial mammals, which consume the mature

Line 342: Orchid bees do not “gather” seeds, so needs to be reworded. 

Line 344: Unclear what “the second” refers to. Better write arboreal mammals consume.

Line 378: Bracket before citations and unnecessary parenthesis at the end

Line 392-393: Zoochory may simply be plesiomorphic so should write as: However, zoochory occurs primarily in…

Line 395: fleshiness not succulence

Line 426: dispersers

Comments on the Quality of English Language

Appropriate

Author Response

Line 46: Unnecessary parenthesis after [4]
R/ Corrected

Line 62: traits
R/ Corrected

Line 66 Parenthesis after [6,13-22]
R/ Corrected

Line 85: respond
R/ Corrected

Line198: conserved, rather than linked
R/ Corrected

Figure 1: The illustration of camel cricket does not really look like crickets. Better be redrawn.
R/ The illustration represents crickets in general, not only camel crickets. We feel it is better to leave it as it is. The illustrations of mammals and birds also represent a general, not specific, animal.

Figure 2: More useful to have subgenus and section names on the tree. Presence/Absence for “2. Vanillin and related compounds” should be Present/Absent.
R/ We changed the figure incorporating the requested changes.

Line 241: Species or taxonomic groups do not “learn”, so the sentence in my opinion can be omitted.
R/ Sentence omitted.

Line 259: shown
R/ Corrected

Line 269 and 270: Unnecessary periods
R/ Corrected

Line 340: terrestrial mammals, which consume the mature
R/ Corrected

Line 342: Orchid bees do not “gather” seeds, so needs to be reworded. 
R/ Corrected

Line 344: Unclear what “the second” refers to. Better write arboreal mammals consume.
R/ We reworded this sentence.

Line 378: Bracket before citations and unnecessary parenthesis at the end
R/ Corrected

Line 392-393: Zoochory may simply be plesiomorphic so should write as: However, zoochory occurs primarily in…
R/ Corrected

Line 395: fleshiness not succulence
R/ Corrected

Line 426: dispersers
R/ Corrected

Reviewer 2 Report

Comments and Suggestions for Authors

This is an excellent overview, with only a few minor corrections.

Line 46: correct (, the

Line 67: please follow the journal formatting.

Line 151: correct Orchidaceae. .

Line 161:  make up comprise Vanilla subgen

Line 270: Dasyproctidae). actively, delete punctuation. 

Author Response

Line 46: correct (, the
R/ Corrected

Line 67: please follow the journal formatting.
R/ Corrected

Line 151: correct Orchidaceae. .
R/ Corrected

Line 161:  make up comprise Vanilla subgen
R/ Corrected

Line 270: Dasyproctidae). actively, delete punctuation. 
R/ Corrected